

# Morphometric and meristic diversity of the species *Androctonus crassicauda* (Olivier, 1807) (Scorpiones: Buthidae) in Saudi Arabia

Abdulaziz R. Alqahtani[1], Ahmed Badry[2], Fahd Mohammed Abd Al Galil[1,3] and Zuhair S. Amr[4]

[1] Department of Biology, College of Science, University of Bisha, Bisha, Saudi Arabia
[2] Department of Zoology, Faculty of Science (Boys), Al-Azhar University, Cairo, Egypt
[3] Department of Biology, Faculty of Applied Sciences, Thamar University, Thamar, Yemen
[4] Department of Biology, Jordan University of Science and Technology, Jordan, Jordan

Corresponding author
Ahmed Badry, ahmed-badry@azhar.edu.eg

## ABSTRACT

Intraspecific molecular and morphological variations among geographically isolated populations are useful for understanding the evolutionary processes, which is considered early stage of allopatric speciation. Also, the knowledge of the regional variation of scorpion venom composition is needed to improve antivenom therapeutic management. *Androctonus crassicauda* (Olivier, 1807) is the most common and medically important species in Arabia and the Middle East. Therefore, this study aimed to investigate the geographic morphological variation among *A.crassicauda* populations, regarding its geographical distribution in unexplored arid regions in Saudi Arabia. Samples were collected and examined morphologically under a dissecting microscope from different four eco-geographical regions. The results of ANOVA and multivariate statistical analyses provide strong evidence of geographical variation. The two populations from OTU3 and OUT4 showed the greatest degree of morphological difference from populations of OUT1 and OUT2. Each OTU3 and OTU4 populations showed significant speciation without overlapping in the two groups, while the remaining overlapped groups comprised two other populations. Several body variables influenced male separation, including carapace posterior width, metasoma 3rd length, and metasoma 2nd length. For females, telson length, metasoma 1st width, and sternite 7th width were highly influential variables. Such variation may suggest the existence of cryptic taxa within A. crassicauda populations in Saudi Arabia. Moreover, metasoma ratios can be used as good indicators in intraspecific variation studies of Scorpions.

## INTRODUCTION

Family Buthidae C. L. Koch, 1837 encompass some of the world's most widely distributed, extant scorpions. It is the largest of scorpion families with 96 genera (*Rein, 2022a*; *Rein, 2022b*), including medically important species (*Polis, 1990*; *Lourenço, 2002*; *Chippaux & Goyffon, 2008*; *Ozkan et al., 2008*). The genus *Androctonus* Ehrenberg, 1828, was introduced
by Ehrenberg in Ehrenbergand Hemprich (1828), belongs family Buthidae, with 30 valid named species (*Rein, 2022a*; *Rein, 2022b*; *Yağmur, 2021*; *Ythier & Lourenço, 2022*). Several studies have detailed with the systematics of this genus (e.g., *Vachon, 1948*; *Vachon, 1952*; *Lourenço, 2005*; *Lourenço, 2008*; *Lourenço & Qi, 2006*; *Lourenço & Qi, 2007*; *Lourenço, Ythier & Leguin, 2009*; *Lourenço, Duhem & Cloudsley-Thompson, 2012*; *Lourenço, Rossi & Sadine, 2015*; *Kovařík & Ahmed, 2013*; *Teruel, Kovařík & Turiel, 2013*; *Rossi, 2015*; *Ythier & Lourenço, 2022*). *A. crassicauda* (Olivier, 1807) is the most medically important species, distributed across Egypt (Sinai) and the Middle East including Iran (*Alqahtani & Badry, 2021*; *Amr et al., 2021*). Scorpion is well-known to have significant regional variation in venom composition (*Devaux et al., 2004*; *Newton et al., 2007*; *Smertenko et al., 2001*), and thus have a different response to antivenom treatment (*Omran & McVean, 2000*). Furthermore, other species may also cause scorpion envenomation more frequently than currently thought (*Goyffon, Dabo & Coulibaly, 2012*). Recently, Three new species of genus *Buthacus* were described in the Levant, bringing the total to seven species by using qualitative and quantitative morphological characters, and multivariate analysis of morphometric data. As well as carapace dimensions, chela shape, metasomal segments proportions, telson vesicle, and aculeus proportions, pectinal tooth counts, pilosity, and density of macrosetae on metasoma and telson, and macrosculpture of the metasomal carinae, were the most informative morphological characters for species identification (*Cain, Gefen & Prendini, 2021*). Thus, the knowledge about the scorpion taxonomy and identification plays a role in significant scorpion envenomation. Therefore, the identification of this species is essential due to its widespread distribution and medical importanace. The present study aims to investigate the morphological variation of its populations in different eco-geographical regions of the country and discusses the taxonomic implications of that structuring.

## MATERIALS & METHODS

### Sampling

A total of 78 specimens of *A. crassicauda* were collected from January 2021 to July 2021, from four eco-geographical regions of Saudi Arabia (Fig. 1, Table 1). *A. crassicauda* samples were grouped in to Opertional Taxonomic Unites (OTUs) based on their ecogeographical regions, including; the North Arabian Desert (OTU1), Central Arabian Desert (OTU2), Southwestern Arabian Escarpment and Highlands (OTU3) and Tehama plain (OTU4) as showen in Table 2. Scorpion sampling was done mainly at night using ultraviolet lights and during the daytime by randomly searching for their hiding places, according to *Williams (1968)* and *Stahnke (1972)*. The collected scorpions were preserved in 95%ethanol as described by *Prendini, Crowe & Wheeler (2003)*.

### Specimen examined and morphological studies

The collected scorpions were maintained and preserved for permanent storage in 70% alcohol. Specimens were examined under a dissecting microscope. Also, six meristic (countable) and 39 morphometric characters were analyzed. The meristic characters were counted as follows; the number of pectineal teeth (right and left), metasoma II ventromedian carinal denticles and metasoma III ventromedian carinal denticles, pedipalp

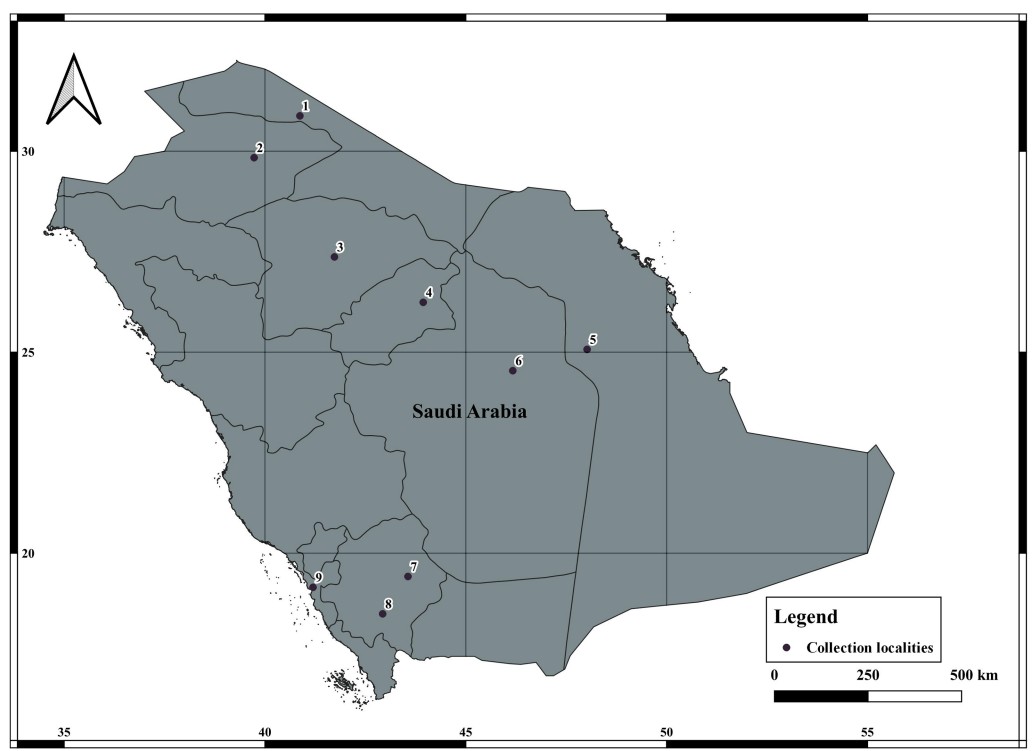

**Figure 1** Collection of localities data of *Androctonus crassicauda* from Saudi Arabia.

**Table 1** Eco-geographical regions, coordinates, number of Morphological specimens of *Androctonus crassicauda* collected from Saudi Arabia.

| No | Location | Region | Ecogeographical region | OTUs | No. of morph. specimens | Lat. | Long. |
|---|---|---|---|---|---|---|---|
| 1 | Arar | Northern Borders Province | | | 7 | 30.88 | 40.87 |
| 2 | Dumah Al Jandal | Al Jowf | **North Arabian Desert** | OTU1 | 7 | 29.84 | 39.73 |
| 3 | Hail | Hail Province | **Central Arabian Desert** | OTU2 | 7 | 27.37 | 41.73 |
| 4 | Buraydah | Al Qassim | | | 7 | 26.241 | 43.94 |
| 5 | Khurais | Eastern Province | | | 6 | 25.07 | 48.02 |
| 6 | Dhurma | Riyadh | **Arabian Sand Desert** | OTU2 | 6 | 24.54 | 46.17 |
| 7 | Tathleeth | Aseer Province | **Southwestern Arabian Escarpment and Highlands** | OTU3 | 9 | 19.42 | 43.56 |
| 8 | Wadi Al Shiq | | | | 8 | 18.49 | 42.93 |
| 9 | East of Al Qunfudhah | Makkah Province | **The Tehama plain** | OTU4 | 21 | 19.150512 | 41.196326 |

denticle sub rows of the movable finger, and pedipalp denticle sub rows of the fixed finger (Files S1 and S2). For measurements, the words length, width, height and depth are abbreviated as L, W, and H, respectively. The morphometric characters were; the total body length, carapace (length, anterior width and posterior width), mesosoma tergite and sternite
**Table 2** List of OTUs of morphological specimens of Androctonus crassicauda collected from Saudi Arabia, based on their ecogeographical regions.

| OTU1 | OTU2 | OTU3 | OTU4 |
|---|---|---|---|
| Comprising specimens from the North Arabian Desert; this ecoregion stretches across northern Saudi Arabia and into Iraq's western desert. Moving east through the Al-Jouf region, a complex of wadis pass through limestone hills towards the Iraqi desert, which is mostly a mixture of gravel and sandy plains interspersed with a few brackish lakes. The climate is hot and dry with mean annual minimum temperatures of 2–15 °C, and maximum temperatures ranging from 25–40 °C. Average annual rainfall is around 50–200 mm. | Comprising specimens from the Central Arabian Desert; the Najd plateau region of central Saudi Arabia and wadis around the Tuwayq escarpment. The elevation range of records is 20–800 m, with the lowest on the east coast, and the highest on the Najd plateau. Most collections were made from vegetated wadis and oases in arid, stony desert in the region around Riyadh, which inhabiting burrows in sandy desert soils. | Southwestern Arabian Escarpment and Highlands; along the chain of mountains running parallel to the Red Sea coast of Saudi Arabia and Yemen (Al Hijaz and Asir mountains). a wide range of elevations (22–2,828 m a.s.l. ranging from coastal plains to the Asir highlands. Specimens were found on rock and gravel substrates in densely vegetated wadis, from coastal plains to mountains. | Tehama plain; along the Red Sea coast of south- western Saudi Arabia. All collections are from low elevation coastal sites (<110 m a.s.l.). The Tihamah plain is a hot environment with daily high temperatures of ca. 43 °C, and 40–60% relative humidity, and the southern sites lie in the zone of coastal fog desert |

7th (length and width), metasomal segments from I-V (length, width and height), telson length, telson vesicle (length and width), pedipalp femur (length and width), pedipalp patella (length and width), pedipalp chela length, pedipalp chela manus (length, width and height), the movable finger length and pectine length (Files S1 and S2). We followed the definitions of all measurements as proposed by *Sissom, Polis & Watt (1990)* and *Cain, Gefen & Prendini (2021)*. Specimens were examined for meristic characters under dissecting microscope and morphometrics were measured with digital calipers in millimeters. All material was deposited at the AUZC, Department of Zoology, Faculty of Science Al-Azhar University, Cairo, Egypt.

## Statistical analysis

The statistical analyses were performed with NCSS 2007 (https://www.ncss.com/download/ncss/updates/). As observed in other scorpions, the results of all morphometric measurements of scorpions were analyzed with one-way ANOVA to determine if there is a significant difference between populations (*Benton, 1992*; *Abdel-Nabi et al., 2004*). Also, the mean ratios that showed proportions of some body measurements were calculated; "chela manus W / total body length, carapace anterior W / posterior W, carapace L / posterior W, chela manus W / L, Chela manus H / length, chela manus L along retroventral carina / movable finger L, metasomal segment I-V (W / L), , metasomal segment I L / segment II L, metasomal segment II L / segment III L, metasomal segment III L / segment IV L, metasomal segment IV L / segment V L, telson vesicle W / metasomal segment V W, telson vesicle H / L, and Sternite VII L / W".

We used multivariate discriminant analysis to assess morphological differences among populations from eco-geographical regions based on morphometric variables (*Fisher, 1936*). Canonical Correlation Coefficient analysis and the Eigenvalues were calculated as proposed by *Vignoli et al. (2005)* and *Olivero, Mattoni & Peretti (2012)*.

## RESULTS

### Morphometric analysis

We statistically analyzed six meristic and 39 morphometric characters and the results indicated that meristic and morphometric measurements of *A. crassicauda* reflected some structuring of the populations. The ANOVA for males from all OTUs localities revealed significant differences in 29 morphological measurement characters and one meristic character, as shown in (Tables 3 and 4). The meristic character showed significant difference includes the number of chela sub row on the movable finger of the pedipalp. While in females, 25 morphometric measurements and one meristic character showed a significant differences, including the number of chela sub row on the movable finger of the pedipalp (Tables 5 and 6). As summarized in Tables 4 & 6, the variation among the number of chela sub row on the movable finger of the pedipalp may help to distinguish between our OUTs. Also, the statistical results indicated a significant differences among OTUs populations in almost all mean body morphometric ratios in males and females (Tables 7 and 8). Samples from OUT3 descibed a significant difference from the other OTUs in some morphometric ratios in females overlapped in males. This ratio were; a relative greater than Metasomal segment I, IV, wider than length (Fig. 2). Notably, the morphometric ratio of OTU2, OTU3 and OTU4 populations were broadly overlapped. Therefore, results indicated that metasoma ratios could be a a good indicator in intraspecific variation studies of *scorpions*.

The multivariate discriminant analysis showed clear discrimination among OTUs populations. In males, a significant separation between the populations collected from OTU3 and OTU4 showed no overlap with any other group was observed. Another overlapped group was collected from OTU1 and OUT2. Discriminant functions analysis revealed that populations close in geographical distance exhibit similar coordinates in the discriminant function analysis (Figs. 3A and 3B). The most discrimination occurred in scores 1 and 2, in which Wilk's Lambda was 0.00001 and 0.0081, respectively (F: 56.9 and 15.1, P, <0.00001) and the Eigenvalue were 601.60 and 39.83 with percentage (93.5% and 6.2% respectively). For the other scores, there was no significant separation. The variables that highly influenced male separation were body variables such as carapace posterior width, metasoma 3rd length and metasoma 2nd length (Table 9). In females, the discriminant analysis differences as regarding in males. Each OTU3 and OTU4 populations showed significant speciation without overlapping in the two groups, while the remaining overlapped groups comprised two other populations. Scores that better explained the variation were score 1 and 2, with Wilk's Lambda 0.00007 and 0.006102, respectively (F: 18.3 and 10.0, $P < 0.00001$) and the Eigenvalue were 86.77 and 15.13 with percentage (78.1% and 13.6% respectively). Female variables that highly influenced population variation were; telson length, metasoma 1st width and sternite 7th width (Table 9). Therefore, our results show a strong predictive of body morphometric variability that suggests three distinct taxa within *A. crassicauda*.

**Table 3 Descriptive and statistical analysis of the morphometric measurements of males of *Androctonus* populations collected from different geographically isolated localities in Saudi Arabia.** The results of one-way ANOVA have been presented in the last column.

| Characters | OTU1 | OTU2 | OTU3 | OUT4 (13) | F-Ratio | P |
|---|---|---|---|---|---|---|
| Total body Length | 77.43 ± 2.65 (6) | 73.42 ± 7.26 (14) | 72.57 ± 5.90 (10) | 77.06 ± 13.06 (8) | 1.1357 | 0.3485[n.s.] |
| Carapace Length | 9.78 ± 0.40 (6) | 8.68 ± 0.89 (14) | 9.00 ± 0.66 (10) | 9.28 ± 0.59 (8) | 3.5789 | 0.0237[**] |
| Carapace anterior W | 6.33 ± 0.46 (6) | 6.25 ± 0.70 (14) | 6.28 ± 0.64 (10) | 6.30 ± 0.38 (8) | 0.0238 | 0.9949[n.s.] |
| Carapace posterior W | 10.49 ± 1.18 (6) | 8.66 ± 0.71 (14) | 9.132 Âś0.67 (10) | 9.3175Âś 0.40 (8) | 8.55 | 0.0002[***] |
| Tergite 7$^{th}$ L | 5.88 ± 0.83 (6) | 5.62 ± 1.15 (14) | 4.976 ± 0.45 (10) | 4.95 ± 0.36 (8) | 2.58 | 0.06[n.s.] |
| Tergite 7$^{th}$ W | 8.74 ± 1.81 (6) | 7.50 ± 1.87 (14) | 8.28 ± 0.74 (10) | 8.8 ± 0.64 (8) | 1.8601 | 0.1549[n.s.] |
| Sternite 7$^{th}$ L | 4.94 ± 1.48 (6) | 5.03 ± 1.39 (14) | 4.22 ± 0.2 (10) | 4.36 ± 0.28 (8) | 1.5320 | 0.2239[n.s.] |
| Sternite 7$^{th}$ W | 8.59 ± 0.90 (6) | 7.67 ± 0.99 (14) | 8.28 ± 0.67 (10) | 8.34 ± 0.81 (8) | 2.1187 | 0.1160[n.s.] |
| Pectine L | 10.03 ± 1.01 (6) | 8.47 ± 0.98 (14) | 9.44 ± 1.29 (10) | 9.57 ± 0.79 (8) | 4.0005 | 0.01528[**] |
| Metasoma 1$^{st}$ L | 7.83 ± 0.48 (6) | 6.67 ± 0.79 (14) | 6.67 ± 0.65 (10) | 7.41 ± 0.48 (8) | 6.1952 | 0.0017[**] |
| Metasoma 1$^{st}$ W | 7.06 ± 0.47 (6) | 5.69 ± 0.86 (14) | 6.31 ± 0.94 (10) | 6.45 ± 0.37 (8) | 4.9490 | 0.0058[***] |
| Metasoma 1$^{st}$ H | 6.1 ± 0.49 (6) | 4.80 ± 0.71 (14) | 5.34 ± 0.76 (10) | 5.61 ± 0.34 (8) | 6.4959 | 0.0013[**] |
| Metasoma 2$^{nd}$ L | 8.47 ± 0.70 (6) | 7.17 ± 0.70 (14) | 7.37 ± 0.59 (10) | 8.04 ± 0.43 (8) | 7.8274 | 0.0004[***] |
| Metasoma 2$^{nd}$ W | 7.61 ± 0.61 (6) | 6.20 ± 0.85 (14) | 7.05 ± 1.18 (10) | 7.33 ± 0.46 (8) | 5.1152 | 0.0049[**] |
| Metasoma 2$^{nd}$ H | 6.80 ± 0.42 (6) | 5.48 ± 0.90 (14) | 5.97 ± 0.76 (10) | 6.21 ± 0.37 (8) | 5.0792 | 0.0051[**] |
| Metasoma 3$^{rd}$ L | 8.86 ± 0.61 (6) | 7.47 ± 0.89 (14) | 7.63 ± 0.49 (10) | 8.39 Âś0.40 (8) | 7.8557 | 0.0004[***] |
| Metasoma 3$^{rd}$ W | 8.32 ± 0.51 (6) | 6.67 ± 1.38 (14) | 7.62 ± 1.33 (10) | 8.02 ± 0.53 (8) | 4.0424 | 0.0146[*] |
| Metasoma 3$^{rd}$ H | 7.80 ± 0.46 (6) | 6.14 ± 1.27 (14) | 6.79 ± 1.10 (10) | 6.98 ± 0.45 (8) | 4.0363 | 0.0147[*] |
| Metasoma 4$^{th}$ L | 9.29 ± 0.43 (6) | 8.04 ± 0.97 (14) | 8.31 ± 0.66 (10) | 8.97 ± 0.40 (8) | 5.4219 | 0.0037[**] |
| Metasoma 4$^{th}$ W | 8.37 ± 0.53 (6) | 6.65 ± 1.42 (14) | 7.73 ± 1.00 (10) | 8.04 ± 0.46 (8) | 5.1346 | 0.0049[**] |
| Metasoma 4$^{th}$ H | 7.79 ± 0.35 (6) | 6.17 ± 1.29 (14) | 6.70 ± 1.09 (10) | 7.00 ± 0.39 (8) | 3.8822 | 0.01727[*] |
| Metasoma 5$^{th}$ L | 11.67 ± 0.56 (6) | 9.89 ± 1.40 (14) | 10.70 ± 0.91 (10) | 10.86 ± 0.49 (8) | 4.5157 | 0.0090[**] |
| Metasoma 5$^{th}$ W | 7.24 ± 0.51 (6) | 5.93 ± 1.15 (14) | 6.55 ± 1.02 (10) | 7.05 ± 0.29 (8) | 4.0559 | 0.0144[*] |
| Metasoma 5$^{th}$ H | 6.09 ± 0.19 (6) | 4.93 ± 0.94 (14) | 5.45 ± 0.82 (10) | 5.44 ± 0.29 (8) | 3.6007 | 0.0231[**] |
| Telson L | 9.40 ± 0.56 (6) | 8.14 ± 0.65 (14) | 8.37 ± 0.73 (10) | 7.83 ± 1.35 (8) | 4.2855 | 0.0114[*] |
| Vesicle L | 5.31 ± 0.35 (6) | 4.63 ± 0.51 (14) | 4.77 ± 0.37 (10) | 4.91 ± 0.21 (8) | 4.1302 | 0.0133[**] |
| Vesicle W | 3.81 ± 0.24 (6) | 3.31 ± 0.30 (14) | 3.50 ± 0.24 (10) | 3.42 ± 0.17 (8) | 5.3017 | 0.0041[**] |
| Vesicle H | 3.18 ± 0.09 (6) | 2.73 ± 0.30 (14) | 3.05 ± 0.42 (10) | 2.88 ± 0.15 (8) | 4.1708 | 0.0128[*] |
| Femur L | 8.17 ± 0.64 (6) | 7.01 ± 0.76 (14) | 7.31 ± 0.68 (10) | 77.07 ± 0.22 (8) | 4.8431 | 0.0065 |
| Femur W | 2.90 ± 0.37 (6) | 2.32 ± 0.27 (14) | 2.59 ± 0.38 (10) | 2.70 ± 0.10 (8) | 5.9400 | 0.0022[**] |
| Femur H | 2.42 ± 0.21 (6) | 1.92 ± 0.24 (14) | 2.13 ± 0.30 (10) | 1.70 ± 0.99 (8) | 2.6060 | 0.0676[n.s.] |
| Patella L | 9.43 ± 0.39 (6) | 8.26 ± 0.70 (14) | 8.60 ± 0.59 (10) | 8.88 ± 0.54 (8) | 5.6372 | 0.00302[**] |
| Patella W | 4.71 ± 0.77 (6) | 3.99 ± 1.43 (14) | 3.62 ± 0.38 (10) | 3.74 ± 0.14 (8) | 1.7794 | 0.16965[n.s.] |
| Patella H | 3.28 ± 0.32 (6) | 2.58 ± 0.27 (14) | 2.73 ± 0.53 (10) | 3.14 ± 0.15 (8) | 8.0119 | 0.00036[***] |
| Pediplap chela L | 16.75 ± 0.70 (6) | 14.79 ± 1.67 (14) | 13.93 ± 2.26 (10) | 14.40 ± 2.84 (8) | 2.5270 | 0.07379[n.s.] |
| Pedipalp chela manus L | 6.08 ± 0.08 (6) | 5.40 ± 0.84 (14) | 5.38 ± 0.63 (10) | 5.86 ± 0.31 (8) | 2.4863 | 0.07718[n.s.] |
| Pedipalp chela manus W | 4.44 ± 0.58 (6) | 3.17 ± 0.80 (14) | 3.48 ± 0.76 (10) | 4.31 ± 0.23 (8) | 7.6159 | 0.00050[***] |
| Pedipalp chela manus H | 5.13 ± 0.58 (6) | 3.87 ± 0.94 (14) | 3.90 ± 0.85 (10) | 4.84 ± 0.28 (8) | 5.8892 | 0.00238[**] |
| Movable finger L | 11.10 ± 0.57 (6) | 9.78 ± 0.92 (14) | 9.97 ± 1.10 (10) | 10.25 ± 0.46 (8) | 3.4749 | 0.02648[*] |

Notes.

Significant difference, ($P < 0.05$: *; $P < 0.01$: **; $P < 0.001$: ***), and non-significant difference (n. s.).

**Table 4  Descriptive and statistical analysis of the meristic characters of males of *Androctnus crassicauda* populations collected from different geographically isolated localities in Saudi Arabia.** The results of one-way ANOVA have been presented in the last column.

| Characters | OTU1 | OTU2 | OTU3 | OUT4 | F-Ratio | P |
|---|---|---|---|---|---|---|
| Chela subrow on movable | 14.83 ± 0.25 (6) | 14.42 ± 0.51 (14) | 14.80 ± 0.42 (10) | 14 ± 0.00 (8) | 7.68 | 0.0004[***] |
| Chela subrow on fixed | 14.33 ± 0.51 (6) | 14 ± 0.00 (14) | 13.6 ± 0.84 (10) | 13.75 ± 0.46 (8) | 2.87 | 0.0501[n.s.] |
| Pecten teeth count Right | 31.16 ± 0.68 | 32 ± 1.35 (14) | 32.8 ± 1.68 (10) | 31.5 ± 0.53 (8) | 2.63 | 0.0655[n.s.] |
| Pecten teeth count Left | 31.50 ± 0.44 (6) | 32.42 ± 1.65 (14) | 32.60 ± 1.57 (10) | 31.50 ± 0.92 (8) | 1.57 | 0.2132[n.s.] |
| Metasoma II Ventromedian Carina denticles | 28.33 ± 1.03 (6) | 29.71 ± 1.63 (14) | 28.6 ± 2.36 (10) | 28.75 ± 2.05 (8) | 1.12 | 0.3530[n.s.] |
| Metasoma III Ventromedian Carina denticles | 30.83 ± 0.68 (6) | 29.85 ± 1.51 (14) | 29.40 ± 1.26 (10) | 30.25 ± 0.46 (8) | 2.01 | 0.1302[n.s.] |

**Notes.**
Significant difference, ($P < 0.05$: *; $P < 0.01$: **; $P < 0.001$: ***), and non-significant difference (n. s.).

# DISCUSSION

The above results indicate some clear geographical variation occurs at the intraspecific level of *Anroctonus crassicauda* populations in both morphometric measurements and meristic characters. Our results showed that *A. crassicauda* populations of the rest of the country are further divided into three sister phylogroups; the first includes OTU1 and OTU2, representing populations in the Northern and Central parts of the country. The second sister phylogroup includes OTU3 representing populations of Southwestern Saudi Arabia. In comparison, the third phylogroup includes OTU4 representing the population of Tehama plain. *Alqahtani et al. (2022a)* and *Alqahtani et al. (2022b)* revealed high genetic diversity and structure among *Androctonus* populations from Saudi Arabia and Iran, based on COI gene. Also, they suggested that "a strong biogeographic barrier between these populations, or that their current proximity is an area of potential secondary contact". In addition, the results of ANOVA and multivariate analyses provide further indication of morphological structure among *A. crassicauda* in males and females, showing that populations at close range in geographical distance matrices (Tables 3 and 4 Fig. 2). This variation may be attributed to the adaptation to gradual geographic changes in climate as morphological differentiation among populations may result from local environmental conditions (*Dillon, 1984*). This adaptation is often expressed as a measurable change in morphological traits. *Abdel-Nabi et al. (2004)* found highly significant differences in most of the morphometric measurements within and among *Sorpio maurus palmatus* populations and referred that these variations are with the influence of environmental factors (altitude, soil nature and climate). Probably these populations were not completely isolated. In our study, scorpion individuals were sampled from populations belonging to different habitats in large-scale isolated eco-geographical regions in Saudi Arabia.

**Table 5 Descriptive and statistical analysis of the meristic measurements of females of Androctonus crassicauda populations collected from different geographically isolated localities in Saudi Arabia.** The results of one-way ANOVA have been presented in the last column.

| Characters | OTU1 | OTU2 (12) | OTU3 | OUT4 | F-Ratio | P |
|---|---|---|---|---|---|---|
| Total body Length | 67.45 ± 5.56 (8) | 77.16 ± 7.49(12) | 79.61 ± 14.04 (7) | 76.15 ± 8.27 (13) | 2.84 | 0.0510[n.s.] |
| Carapace Length | 8.32 ± 0.48 (8) | 9.23 ± 1.37(12) | 9.84 ± 1.73 (7) | 9.54 ± 1.00 (13) | 2.38 | 0.0851[n.s.] |
| Carapace anterior W | 5.44 ± 0.36 (8) | 6.36 ± 1.02(12) | 6.63 ± 1.55 (7) | 6.24 ± 0.69 (13) | 2.27 | 0.0968[n.s.] |
| Carapace posterior W | 8.60 ± 1.14 (8) | 9.49 ± 1.60(12) | 9.92 ± 1.62 (7) | 9.57 ± 0.93 (13) | 1.39 | 0.2591[n.s.] |
| Tergite 7$^{th}$ L | 5.16 ± 0.80 (8) | 5.17 ± 0.43(12) | 5.75 ± 1.08 (7) | 5.02 ± 0.75 (13) | 1.49 | 0.2320[n.s.] |
| Tergite 7$^{th}$ W | 7.98 ± 0.36 (8) | 8.55 ± 0.96(12) | 9.32 ± 1.60 (7) | 9.21 ± 1.14 (13) | 2.86 | 0.0499[*] |
| Sternite 7$^{th}$ L | 4.17 ± 0.69 (8) | 4.54 ± 0.61(12) | 4.57 ± 0.78 (7) | 4.33 ± 0.56 (13) | 0.73 | 0.5380[n.s.] |
| Sternite 7$^{th}$ W | 7.19 ± 0.64 (8) | 8.40 ± 1.00(12) | 9.38 ± 1.76 (7) | 9.12 ± 1.21 (13) | 5.71 | 0.0026[**] |
| Pectine L | 6.76 ± 0.54 (8) | 7.03 ± 1.24(12) | 8.15 ± 1.04 (7) | 7.96 ± 0.59 (13) | 5.13 | 0.0046[*] |
| Metasoma 1$^{st}$ L | 6.15 ± 0.46 (8) | 6.87 ± 0.86(12) | 7.11 ± 1.08 (7) | 7.13 ± 0.60 (13) | 3.04 | 0.0409[*] |
| Metasoma 1$^{st}$ W | 5.15 ± 0.40 (8) | 5.67 ± 0.61(12) | 6.72 ± 1.26 (7) | 6.00 ± 0.61 (13) | 6.06 | 0.0018[**] |
| Metasoma 1$^{st}$ H | 4.65 ± 0.38 (8) | 5.13 ± 0.64(12) | 5.55 ± 0.92 (7) | 4.92 ± 0.79 (13) | 2.17 | 0.1083[n.s.] |
| Metasoma 2$^{nd}$ L | 6.68 ± 0.57 (8) | 7.24 ± 0.92(12) | 7.95 ± 1.18 (7) | 7.68 ± 0.62 (13) | 3.67 | 0.0208[*] |
| Metasoma 2$^{nd}$ W | 5.56 ± 0.44 (8) | 6.21 ± 1.07(12) | 7.12 ± 1.48 (7) | 6.59 ± 0.79 (13) | 3.51 | 0.0246[*] |
| Metasoma 2$^{nd}$ H | 4.84 ± 0.42 (8) | 5.45 ± 0.79(12) | 6.14 ± 1.26 (7) | 5.69 ± 0.65 (13) | 3.58 | 0.0230[*] |
| Metasoma 3$^{rd}$ L | 6.90 ± 0.53 (8) | 7.54 ± 0.93(12) | 8.15 ± 1.31 (7) | 7.92 ± 0.69 (13) | 3.19 | 0.0347[*] |
| Metasoma 3$^{rd}$ W | 5.83 ± 0.64 (8) | 6.66 ± 1.17(12) | 7.71 ± 1.68 (7) | 7.00 ± 0.89 (13) | 3.82 | 0.0178[*] |
| Metasoma 3$^{rd}$ H | 5.29 ± 0.44 (8) | 6.28 ± 1.15 (12) | 7.05 ± 1.48 (7) | 6.36 ± 0.89 (13) | 3.69 | 0.0204[*] |
| Metasoma 4$^{th}$ L | 7.40 ± 0.62 (8) | 8.20 ± 1.11(12) | 8.68 ± 1.27 (7) | 8.55 ± 0.71 (13) | 3.09 | 0.0387[*] |
| Metasoma 4$^{th}$ W | 5.68 ± 0.70 (8) | 6.49 ± 1.10(12) | 7.62 ± 1.31 (7) | 6.96 ± 0.92 (13) | 4.98 | 0.0053[**] |
| Metasoma 4$^{th}$ H | 5.18 ± 0.51 (8) | 6.13 ± 1.07(12) | 7.28 ± 2.09 (7) | 6.25 ± 0.91 (13) | 3.90 | 0.0163[*] |
| Metasoma 5$^{th}$ L | 9.03 ± 0.71 (8) | 10.18 ± 1.67(12) | 11.22 ± 1.73 (7) | 10.49 ± 1.11 (13) | 3.42 | 0.0272[*] |
| Metasoma 5$^{th}$ W | 5.14 ± 0.59 (8) | 5.77 ± 0.92(12) | 6.72 ± 1.25 (7) | 6.04 ± 0.75 (13) | 4.18 | 0.0122[*] |
| Metasoma 5$^{th}$ H | 4.26 ± 0.43 (8) | 4.79 ± 0.79(12) | 5.56 ± 1.09 (7) | 4.99 ± 0.67 (13) | 3.77 | 0.0187[*] |
| Telson L | 7.72 ± 0.57 (8) | 8.16 ± 0.82(12) | 9.20 ± 1.08 (7) | 8.97 ± 0.76 (13) | 6.30 | 0.0015[**] |
| Vesicle L | 4.58 ± 0.42 (8) | 5.33 ± 1.16(12) | 5.05 ± 0.60 (7) | 5.34 ± 0.91 (13) | 1.48 | 0.2351[n.s.] |
| Vesicle W | 3.25 ± 0.32 (8) | 3.37 ± 0.63(12) | 4.00 ± 0.68 (7) | 3.59 ± 0.40 (13) | 3.06 | 0.0401[*] |
| Vesicle H | 2.85 ± 0.36 (8) | 2.73 ± 0.50(12) | 3.19 ± 0.52 (7) | 3.07 ± 0.34 (13) | 2.13 | 0.1123[n.s.] |
| Femur L | 6.62 ± 0.58 (8) | 7.36 ± 1.03(12) | 7.77 ± 1.27 (7) | 7.15 ± 0.88 (13) | 1.92 | 0.1430[n.s.] |
| Femur W | 2.39 ± 0.15 (8) | 2.66 ± 0.48(12) | 2.74 ± 0.44 (7) | 2.66 ± 0.35 (13) | 1.24 | 0.3077[n.s.] |
| Femur H | 1.86 ± 0.14 (8) | 3.79 ± 3.69(12) | 2.18 ± 0.44 (7) | 2.19 ± 0.25 (13) | 1.95 | 0.1389[n.s.] |
| Patella L | 7.70 ± 0.59 (8) | 8.50 ± 1.05(12) | 9.37 ± 1.46 (7) | 8.69 ± 0.71 (13) | 3.87 | 0.0169[*] |
| Patella W | 3.12 ± 0.21 (8) | 3.53 ± 0.58(12) | 3.89 ± 0.72 (7) | 3.70 ± 0.51 (13) | 2.97 | 0.0442[*] |
| Patella H | 2.51 ± 0.28 (8) | 2.77 ± 0.48(12) | 3.32 ± 0.68 (7) | 3.07 ± 0.46 (13) | 4.21 | 0.0118[*] |
| Pediplap chela L | 13.75 ± 1.26 (8) | 15.26 ± 2.08(12) | 16.39 ± 2.62 (7) | 15.58 ± 1.61 (13) | 2.58 | 0.0682[n.s.] |
| Pedipalp chela manus L | 4.98 ± 0.51 (8) | 5.30 ± 0.92(12) | 5.88 ± 1.51 (7) | 5.70 ± 0.80 (13) | 1.53 | 0.2231[n.s.] |
| Pedipalp chela manus W | 2.65 ± 0.43 (8) | 3.31 ± 0.75(12) | 3.71 ± 0.95 (7) | 3.65 ± 0.65 (13) | 3.93 | 0.0158[*] |
| Pedipalp chela manus H | 3.21 ± 0.48 (8) | 3.84 ± 0.88(12) | 4.38 ± 1.03 (7) | 4.45 ± 0.84 (13) | 4.24 | 0.0114[*] |
| Movable finger L | 8.91 ± 1.22 (8) | 10.11 ± 1.17(12) | 10.91 ± 1.81 (7) | 10.25 ± 0.79 (13) | 3.66 | 0.0211[*] |

**Notes.**
Significant difference, ($P < 0.05$: *; $P < 0.01$: **; $P < 0.001$: ***).

**Table 6 Descriptive and statistical analysis of the meristic characters of females of *Androctnus crassicauda* populations collected from different geographically isolated localities in Saudi Arabia.** The results of one-way ANOVA have been presented in the last column.

| Characters | OTU1 | OTU2 | OTU3 | OUT4 | F-Ratio | P |
|---|---|---|---|---|---|---|
| Chela subrow on movable | $14.62 \pm 0.51$ (8) | $15.16 \pm 0.38$ (12) | $14.71 \pm 0.48$ (7) | $14.30 \pm 0.75$ (13) | 4.76 | $0.0067$[***] |
| Chela subrow on fixed | $13.87 \pm 0.64$ (8) | $14.33 \pm 0.77$ (12) | $14.00 \pm 0.00$ (7) | $13.76 \pm 0.43$ (13) | 2.17 | $0.1075$[n.s.] |
| Pecten teeth count Right | $25.33 \pm 1.03$ (6) | $25.50 \pm 1.78$ (12) | $26.42 \pm 2.93$ (7) | $25.41 \pm 0.99$ (12) | 0.61 | $0.6113$[n.s.] |
| Pecten teeth count Left | $25.33 \pm 1.86$ (6) | $25.50 \pm 1.44$ (12) | $27.57 \pm 2.43$ (7) | $25.58 \pm 1.88$ (12) | 2.35 | $0.0899$[n.s.] |
| Metasoma II Ventromedian Carina denticles | $26.87 \pm 1.45$ (8) | $28.83 \pm 2.85$ (12) | $28.42 \pm 2.14$ (7) | $30.23 \pm 3.56$ (13) | 2.41 | $0.0826$[n.s.] |
| Metasoma III Ventromedian Carina denticles | $27.25 \pm 2.43$ (8) | $29.66 \pm 2.30$ (12) | $28.71 \pm 4.30$ (7) | $30.53 \pm 2.10$ (13) | 2.60 | $0.0664$[n.s.] |

**Notes.**
Significant difference, ($P < 0.05$: *; $P < 0.01$: **; $P < 0.001$: ***).

Accordingly, four new species of the genus Leiurus were described by *Lowe, Yağmur & Kovařík (2014)* based on quantitative and qualitative morphological variations in Saudi Arabia. *Sarhan et al. (2020)* revealed that the genetic distance between *Leiurus quinquestriatus* populations from Egypt might be two distinct species in the North Africa and Asian part of Egypt (*Sinai Peninsula*). Also, *Omran & McVean (2000)* revealed differences in the venom components. Their physiological effectiveness has been exhibited in the venom of the scorpion *Leiurus quinquestriatus* collected from two different geographic regions in Egypt. Specifically, some efforts have been conducted on *A. crassicauda* based on mitochondrial DNA markers in Turkey and Iran (*Ozkan, Ahmet & Zafer, 2010*; *Toprak, Parmaksiz & Aslan, 2019*; *Jafari et al., 2020*). These findings support the description of a new species of the *Androctonus* from Turkey (*Yağmur, 2021*).

There are geographical formations separating them and the morphological divergence between populations, it appears to be the populations in the North Arabian Desert (OTU1) are distinct from those of the Central Arabian Desert (OTU2), Southwestern Arabian Escarpments and Heights (OUT3), and the Tehama plain (OTU4). Thus, morphological results indicate strong predictive of body proportion variation and suggest the existence of distinct taxa within *A. crassicauda* in Saudi Arabia. The association of different morphological types that were found within *A. crassicauda*, is probably due to the geographical position of Arabia in the middle of the old-world continents (Fig. 1). The local environmental conditions may affect scorpion morphology more than genetic structure among populations (*Yamashita & Polls, 1995*). *Levy & Amitai (1980)*, revealed that these vitiations might be caused by a physical barrier prevents gene flow between Asian and African populations of Egypt. The geographical features might contribute to increasing scorpion diversification in association with long-term geomorphological and climatic processes reported in different taxa (e.g., *Sanmartín, 2003*; *Lourenço & Rossi, 2016*;

**Table 7 Variation in adult morphometric ratios of males among different populations of *Androctonus crassicauda* collected from different geographically isolated localities in Saudi Arabia (mean ± SD, sample sizes in parentheses).** The results of one-way ANOVA have been presented in the last column.

| Characters | OTU1 | OTU2 | OTU3 | OUT4 | F-Ratio | P |
|---|---|---|---|---|---|---|
| Chela manus width/Total body length 2 | 0.05 ± 0.00 (2) | 0.04 ± 0.006 (14) | 0.04 ± 0.008 (10) | 0.05 ± 0.0005 (4) | 4.58 | 0.0105[*] |
| Carapace anterior width / posterior width | 0.29 ± 0.35 (6) | 0.72 ± 0.03 (14) | 0.68 ± 0.05 (10) | 0.36 ± 0.32 (8) | 9.80 | 0.0000[***] |
| Carapace length / posterior width | 0.70 ± 0.25 (6) | 1.00 ± 0.04 (14) | 0.98 ± 0.05(10) | 0.83 ± 0.16 (8) | 9.05 | 0.0001[***] |
| Chela manus width / length | 0.79 ± 0.14 (6) | 0.58 ± 0.05 (14) | 0.64 ± 0.07(10) | 0.86 ± 0.14 (8) | 17.04 | <0.00001[***] |
| Chela manus height / length | 0.77 ± 0.03(6) | 0.70 ± 0.06 (14) | 0.72 ± 0.11(10) | 0.78 ± 0.05 (8) | 2.05 | 0.1247[n.s.] |
| Chela manus length along retroventral carina / movable finger length | 0.78 ± 0.17 (6) | 0.54 ± 0.03 (14) | 0.54 ± 0.04 (10) | 0.69 ± 0.13 (8) | 12.42 | <0.00001[***] |
| Metasomal segment I width / length | 0.65 ± 0.17 (6) | 0.85 ± 0.04 (14) | 0.95 ± 0.16 (10) | 0.71 ± 0.16 (8) | 8.17 | 0.0003[***] |
| Metasomal segment II width / length | 0.90 ± 0.01 (6) | 0.86 ± 0.04 (14) | 0.95 ± 0.13(10) | 0.89 ± 0.04 (8) | 2.93 | 0.0473[*] |
| Metasomal segment III width / length | 0.91 ± 0.02 (6) | 0.88 ± 0.08 (14) | 0.99 ± 0.13 (10) | 0.92 ± 0.03 (8) | 3.04 | 0.0417[*] |
| Metasomal segment IV width / length | 0.91 ± 0.04 (6) | 0.82 ± 0.08 (14) | 0.92 ± 0.05 (10) | 0.93 ± 0.04 (8) | 7.97 | 0.0003[***] |
| Metasomal segment V width / length | 0.81 ± 0.15 (6) | 0.59 ± 0.06 (14) | 0.61 ± 0.05 (10) | 0.77 ± 0.12 (8) | 10.81 | 0.0000[***] |
| Metasomal segment I length / segment II length | 0.73 ± 0.17 (6) | 0.92 ± 0.025 (14) | 0.90 ± 0.04 (10) | 0.78 ± 0.14 (8) | 7.58 | 0.0005[***] |
| Metasomal segment II length / segment III length | 0.91 ± 0.01 (6) | 0.96 ± 0.02 (14) | 0.96 ± 0.03 (10) | 0.93 ± 0.01 (8) | 7.61 | 0.0005[***] |
| Metasomal segment III length / segment IV length | 0.95 ± 0.01 (6) | 0.92 ± 0.01 (14) | 0.91 ± 0.01 (10) | 0.95 ± 0.01 (8) | 8.45 | 0.0002[***] |
| Metasomal segment IV length / segment V length | 0.91 ± 0.08 (6) | 0.81 ± 0.06 (14) | 0.77 ± 0.008 (10) | 0.87 ± 0.06 (8) | 8.91 | 0.0001[***] |
| Telson vesicle width metasomal segment V width | 0.70 ± 0.13 (6) | 0.56 ± 0.05 (14) | 0.54 ± 0.05 (10) | 0.65 ± 0.18 (8) | 3.91 | 0.0167[**] |
| Telson vesicle height / length | 0.56 ± 0.05 (6) | 0.59 ± 0.03 (14) | 0.64 ± 0.08 (10) | 0.55 ± 0.08 (8) | 3.19 | 0.0355[**] |
| Sternite VII length / width | 0.69 ± 0.17 (6) | 0.67 ± 0.23 (14) | 0.51 ± 0.02 (10) | 0.54 ± 0.02 (8) | 2.95 | 0.0461[**] |

**Notes.**
Significant difference, ($P < 0.05$: *; $P < 0.01$: **; $P < 0.001$: ***).

*Saleh et al., 2018*; *Alqahtani & Badry, 2020*). Accordingly, *Prendini (2001)* referred to local populations might diverge on small spatial scales as the gene flow is limited across barriers of unsuitable habitat.

**Table 8 Variation in adult morphometric ratios of females among different populations of *Androctonus crassicauda* collected from different geographically isolated localities in Saudi Arabia (mean ± SD, sample sizes in parentheses).** The results of one-way ANOVA have been presented in the last column.

| Characters | OTU1 | OTU2 | OTU3 | OUT4 | F-Ratio | P |
|---|---|---|---|---|---|---|
| Chela manus width/Total body length | 0.03 ± 0.00 (6) | 0.04 ± 0.00 (12) | 0.04 ± 0.00 (6) | 0.04 ± 0.00 (8) | 7.13 | 0.0010[***] |
| Carapace anterior width / Posterior width | 0.52 ± 0.29 (8) | 0.67 ± 0.03 (12) | 0.57 ± 0.24 (7) | 0.42 ± 0.3 (13) | 2.09 | 0.1178[n.s.] |
| Carapace length / posterior width | 0.89 ± 0.22 (8) | 0.97 ± 0.04 (12) | 0.93 ± 0.13 (7) | 0.85 ± 0.19 (13) | 1.21 | 0.3184[n.s.] |
| Chela manus width / length | 0.60 ± 0.17 (8) | 0.52 ± 0.22 (12) | 0.70 ± 0.15 (7) | 0.78 ± 0.18 (13) | 4.10 | 0.0132[*] |
| Chela manus height / length | 0.60 ± 0.06(8) | 0.60 ± 0.25(12) | 0.73 ± 0.12(7) | 0.73 ± 0.10(13) | 2.03 | 0.1266[n.s.] |
| Chela manus length along retroventral carina / movable finger length | 0.61 ± 0.10 (8) | 1.37 ± 2.00 (12) | 0.53 ± 0.06 (7) | 0.63 ± 0.08 (12) | 1.35 | 0.2732[n.s.] |
| Metasomal segment I width / length | 0.75 ± 0.14 (8) | 0.82 ± 0.01 (12) | 0.90 ± 0.16 (7) | 0.72 ± 0.17 (12) | 3.05 | 0.0406[*] |
| Metasomal segment II width / length | 0.84 ± 0.02 (8) | 0.85 ± 0.04 (12) | 0.90 ± 0.05 (7) | 0.85 ± 0.04 (13) | 2.60 | 0.0665[n.s.] |
| Metasomal segment III width / length | 0.83 ± 0.04 (8) | 0.87 ± 0.05 (12) | 0.92 ± 0.07 (7) | 0.87 ± 0.05 (13) | 2.99 | 0.0432[*] |
| Metasomal segment IV width / length | 0.78 ± 0.05 (8) | 0.78 ± 0.03 (12) | 0.88 ± 0.01 (7) | 0.83 ± 0.03 (13) | 11.48 | 0.0000[*] |
| Metasomal segment V width / length | 0.60 ± 0.11 (8) | 0.56 ± 0.04 (12) | 0.63 ± 0.08 (7) | 0.65 ± 0.10 (13) | 2.00 | 0.1307[n.s.] |
| Metasomal segment I length / segment II length | 0.85 ± 0.13 (8) | 0.94 ± 0.01 (12) | 0.85 ± 0.14 (7) | 0.79 ± 0.18 (13) | 2.77 | 0.0552[n.s.] |
| Metasomal segment II length / segment III length | 0.95 ± 0.02 (8) | 0.96 ± 0.01 (12) | 0.95 ± 0.03 (7) | 0.94 ± 0.02 (13) | 0.79 | 0.5036[n.s.] |
| Metasomal segment III length / segment IV length | 0.94 ± 0.02 (8) | 0.92 ± 0.01 (12) | 0.95 ± 0.02 (7) | 0.94 ± 0.03 (13) | 2.96 | 0.0449[*] |
| Metasomal segment IV length / segment V length | 0.82 ± 0.06 (8) | 0.81 ± 0.06 (12) | 0.79 ± 0.05 (7) | 0.85 ± 0.06 (13) | 1.54 | 0.2197[n.s.] |
| Telson vesicle width / metasomal segment V width | 0.70 ± 0.13 (8) | 0.58 ± 0.07 (12) | 0.62 ± 0.07 (7) | 0.68 ± 0.13 (13) | 2.66 | 0.0624[n.s.] |
| Telson vesicle height / length | 0.62 ± 0.03 (8) | 0.53 ± 0.13 (12) | 0.63 ± 0.02 (7) | 0.59 ± 0.03 (13) | 2.90 | 0.0479[*] |
| Sternite VII length / width | 0.46 ± 0.28 (8) | 0.54 ± 0.07 (12) | 0.49 ± 0.02 (7) | 0.50 ± 0.11 (13) | 0.52 | 0.6654[n.s.] |

**Notes.**

Significant difference, ($P < 0.05$: *; $P < 0.01$: **; $P < 0.001$: ***).

# CONCLUSION

*A. crassicauda* populations in Saudi Arabia exhibit clear morphological structuring. Such discrete populations can be readily distinguished based on a number of morphologic characters, particularly size of metasomal segment ratios. It may be due to a physical or ecological barrier causing restriction of gene flow between *A. crassicauda* populations. Also,

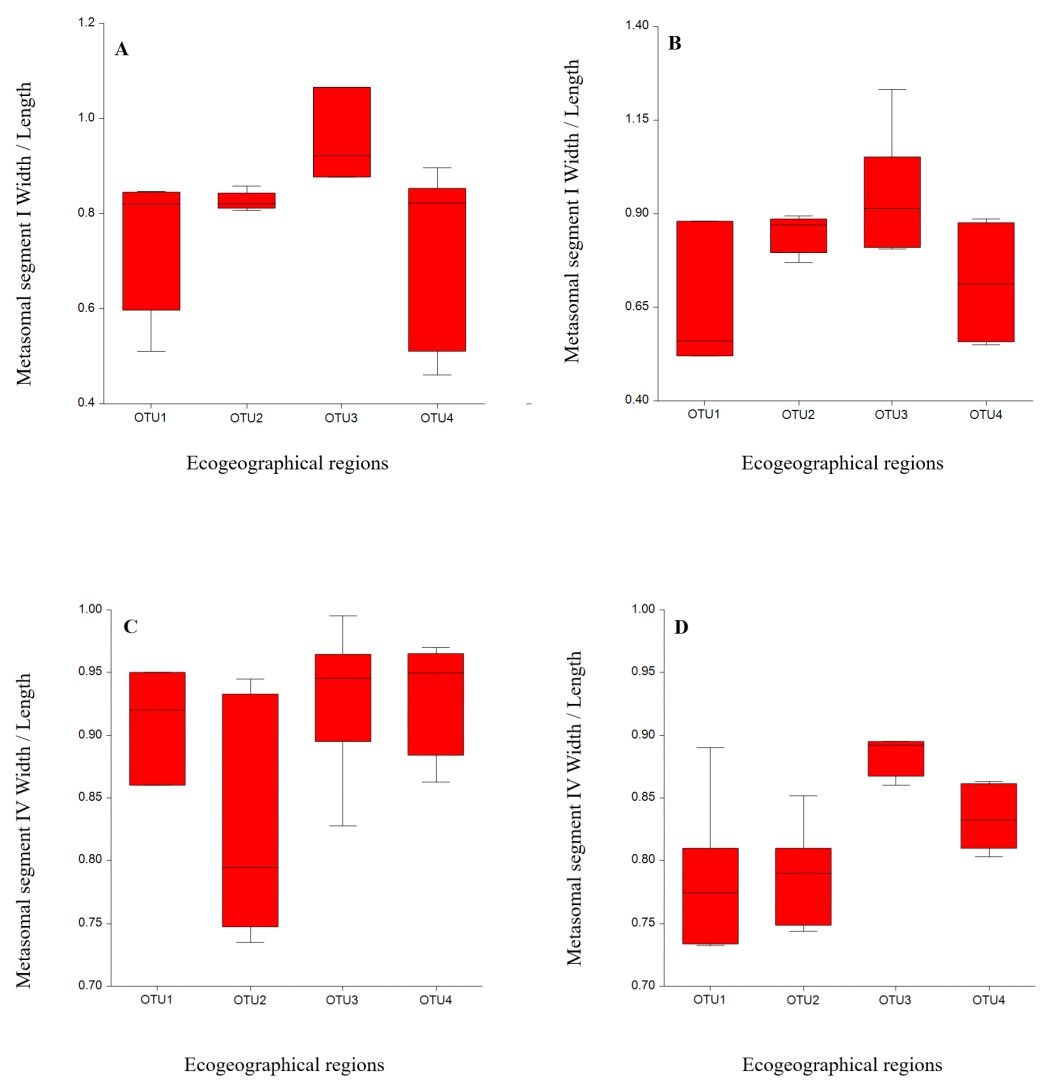

**Figure 2** **Variation by mean of box plots of the morphometric ratio between *Androctonus crassicauda* populations: Metasoma I W/L (A, B); metasoma IV W/L (C, D) for each male and female, respectively.**

the geographical features may significant increasing scorpion propensity that promote diversification with long-term fluctuations such as geomorphological evolution and climatic cycles. In addition, The association of different morphological types that were found within *A. crassicauda*, is probably due to the geographical position of Saudi Arabia in the middle of the old-world continents.

**Table 9** Influence of variables considered in the Canonical Discriminant Analysis for group separation in females and males of *Androctonus crassicauda* populations from different geographical isolated localities in Saudi Arabia.

| | | | Variable Influence Section | | |
|---|---|---|---|---|---|
| **Variable** | **F-Value** | **P** | **Variable** | **F-Value** | **P** |
| | **Males** | | | **Females** | |
| Carapace Posterior W | 8.55 | 0.0002*** | Telson L | 6.31 | 0.0015** |
| Metasoma 3rd L | 7.86 | 0.0004*** | Metasoma 1st W | 6.06 | 0.0018** |
| Metasoma 2nd L | 7.83 | 0.0004*** | Sternite 7th W | 5.71 | 0.0026** |
| Metasoma 1st H | 6.5 | 0.0013** | Pectine L | 5.13 | 0.0046** |
| Metasoma 1st L | 6.2 | 0.0017** | Metasoma 4th W | 4.99 | 0.0053** |
| Metasoma 4th L | 5.42 | 0.0037** | Patella H | 4.21 | 0.0118* |
| Metasoma 4th W | 5.13 | 0.0048** | Metasoma 5th W | 4.18 | 0.0122* |
| Metasoma 2nd W | 5.12 | 0.0049** | Metasoma 4th H | 3.91 | 0.0163* |
| Metasoma 2nd H | 5.08 | 0.0051** | Patella L | 3.87 | 0.0169* |
| Metasoma 1st W | 4.95 | 0.0058** | Metasoma 3rd W | 3.82 | 0.0178* |
| Metasoma 3rd W | 4.04 | 0.0146* | Metasoma 5th H | 3.78 | 0.0187* |
| Metasoma 3rd H | 4.04 | 0.0147* | Metasoma 3rd H | 3.69 | 0.0204* |
| Pectine L | 4 | 0.0152* | Metasoma 2nd L | 3.67 | 0.0208* |
| Metasoma 4th H | 3.88 | 0.0172* | Metasoma 2nd H | 3.58 | 0.0230* |
| Carapace L | 3.58 | 0.0237* | Metasoma 5th L | 3.43 | 0.0272* |
| | | | Metasoma 3rd L | 3.2 | 0.0347* |
| | | | Metasoma 4th L | 3.1 | 0.0387* |
| | | | Vesicle W | 3.07 | 0.0401* |
| | | | Metasoma 1st L | 3.05 | 0.0409* |
| | | | Patella W | 2.98 | 0.0442* |
| | | | Tergite 7th W | 2.87 | 0.0499* |

**Notes.**
Significant difference, ($P < 0.05$: *; $P < 0.01$: **; $P < 0.001$: ***).

### Funding

This research and the APC was funded by the Deputyship for Research and Innovation, Ministry of Education in Saudi Arabia, grant number (UB-46-1442). The funders had no role in study design, data collection and analysis, decision to publish, or preparation of the manuscript.

### Grant Disclosures

The following grant information was disclosed by the authors:
Deputyship for Research and Innovation, Ministry of Education in Saudi Arabia: UB-46-1442.

### Competing Interests

The authors declare there are no competing interests.

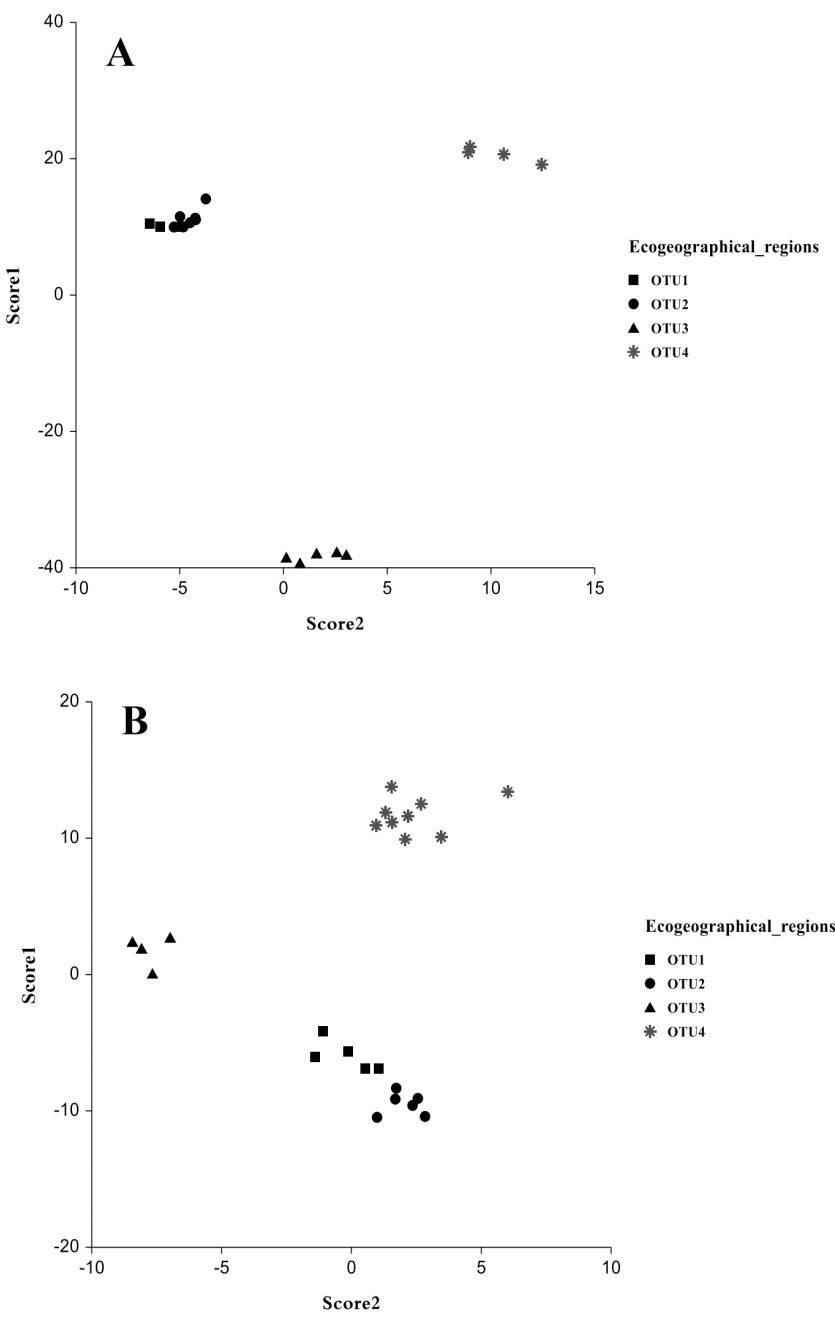

**Figure 3** Discrimination of populations of *Androctonus crassicauda* in the space of Score 1 and 2 as canonical discriminant analysis. Male (A) and female (B).

## Author Contributions

- Abdulaziz R. Alqahtani conceived and designed the experiments, performed the experiments, analyzed the data, prepared figures and/or tables, authored or reviewed drafts of the article, and approved the final draft.

- Ahmed Badry conceived and designed the experiments, performed the experiments, analyzed the data, prepared figures and/or tables, authored or reviewed drafts of the article, and approved the final draft.
- Fahd Mohammed Abd Al Galil conceived and designed the experiments, performed the experiments, analyzed the data, prepared figures and/or tables, authored or reviewed drafts of the article, and approved the final draft.
- Zuhair S. Amr conceived and designed the experiments, performed the experiments, analyzed the data, prepared figures and/or tables, authored or reviewed drafts of the article, and approved the final draft.

## Data Availability

The raw data is available in the Supplemental Files.

## Supplemental Information

Supplemental information for this article can be found online at http://dx.doi.org/10.7717/peerj.14198#supplemental-information.

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
