# Peer review of "Morphometric and meristic diversity of the species Androctonus crassicauda (Olivier, 1807) (Scorpiones: Buthidae) in Saudi Arabia"

_PeerJ, doi:10.7717/peerj.14198_

## Round 0.1 · original submission · Major Revisions

Dear Authors

Thank you for your submission in PeerJ. After the review evaluation, we have reached the Major Decision of your submission. Comments of reviewers 2 and 3 are more significant to incorporate in the revised version.

Best


Reviewer 1 ·

Basic reporting

No comment.

Experimental design

No comment.

Validity of the findings

No comment.

Additional comments

This paper reports results of morphometric and meristic investigations on Androctonus crassicauda in Saudi Arabia. The authors researched well the four populations in Saudi Arabia and justified their results statistically. It is appropriate to publish.

Annotated reviews are not available for download in order to protect the identity of reviewers who chose to remain anonymous.

Reviewer 2 ·

Basic reporting

Dear Reviewers
Thank you for your submission "Morphometric and meristic diversity of the species Androctonus crassicauda (Olivier, 1807) (Scorpiones: Buthidae) in Saudi Arabia" in PeerJ. The overall article is interesting and scientifically sound but many point still required to incorporate.
The introduction section is weak in its present form. Author must take care of its editing and addition of more text. I have suggested a research article which may be helpful for the authors. https://www.sciencedirect.com/science/article/pii/S1055790321001457
The objectives of the study should be extended and well descriptive.
Other minor comments are in the attached file.

Experimental design

Line 57-63: Which solution as used to preserve these collected specimens?? Before and after identification. Write about the study sites, their ecological condition/environmental variables. If author can make the table containing the data of vegetation, temperature, humidity, rainfall and distance of the studied sites from each other.
Line 65: It should be included in the above heading, also cite which methods was used for this preservation.
Line 77: and what about specifically?
Line 94-97: Include this section in the statistical analysis section.

Validity of the findings

This section is of good interest. Authors reveal the data in good manner.
One suggestion is that, authors should comprehensively describe the tables data with their results value. This will make the results more appropriate.

Some minor comments are in the attached file.

Additional comments

Discussion is not well structure and relevant to the content.
Please avoid to be straight forward in writing the results of other. Authors should discuss the findings of their studies with explanatory reasons. Please read some relevant articles for clearing some view points.
Authors have added other results only, compare to their own findings.
Also delete redundant results portion.
Add conclusion section.
Modify abstract majorly.

Annotated reviews are not available for download in order to protect the identity of reviewers who chose to remain anonymous.

Reviewer 3 ·

Basic reporting

It is an interesting study about the scorpions meanwhile, title is also appealing and having significance in ecological study. But, this manuscript have major loop holes according to the research. Both data and write up are very insufficient, ambiguous and unprofessional English have been used whereas, irrelevant things also have been added. It is unable to fulfil the requirements of international audience. As well as introduction part is concerned. This part need more detail. Background of the introduction is not well mentioned. Please make it stronger according to the study. Add few more paragraphs about the scorpions morphology, types, their habitats, habitats type, significance according to their region. Methodology part also need more description about sampling method, Sampling area, Duration of sampling And few more if you found suitable according to the manuscript. Here, few changes in the manuscript are recommended, please consider it.
L37, Write the scientist name and year within brackets
L38; Replace noun with pronoun; Replace Buthidae >it
L40, L41; remove these lines, unnecessary lines
L45; Remove “Of these”
L46; Remove “Belong”. Use another appropriate word
L49; Add few names about “ other countries”
L50; Grammatically weak line. Improve it
L51, L52; Meaningless wording So, Rewrite these lines
L55; Grammatically incorrect.
L58; Replace Between >From
L58; Please also mention year with January
L61; Mention the exact sampling method. Sampling area, Duration of sampling.

Experimental design

Research data seems sufficient but this part of manuscript required few more additions to make the research more authentic and appealing. Research questions need to more descriptive . Described research method also required more detail about the sampling way such as the exact area of all sampling sites, sampling time period, which exact way was used for the sampling? eco-geography of all sampling habitats. Add Figures of the morphometry of scorpions to improve the methodology part.

Validity of the findings

All underlying data is statistically sound and robust. Conclusion is well stated and also support to the results and original research.

Additional comments

It is a novel study and authors have made good effort in the research. but this manuscript needs Major Changes to fulfil the PeerJ criteria.

---

## Round 0.2 · accepted · Accept

Dear Authors

Thank you for your submission. I am please to announce that your manuscript has been accented for publication. Reviewers have confirmed, authors have made all the changes in manuscript that were suggested. Based on the comments of reviewer I accept the submission.

Best

Reviewer 2 ·

Basic reporting

Improvements has been done as suggested.

Experimental design

No comments

Validity of the findings

No comments

Reviewer 3 ·

Basic reporting

Manuscript is very well and in better form after all recommended changes. I think, there is no need to change anymore. sufficient background knowledge have been provided in detail. .

Experimental design

This part has also been improved according to the mentioned changes and now, it sounds good.

Validity of the findings

N/A

Additional comments

Manuscript is fulfilling all requirements of the Journal.